# Work–Life Balance: Weighing the Importance of Work–Family and Work–Health Balance

**DOI:** 10.3390/ijerph17030907

**Published:** 2020-02-01

**Authors:** Andrea Gragnano, Silvia Simbula, Massimo Miglioretti

**Affiliations:** Department of Psychology, University of Milano-Bicocca, 20126 Milan, Italy

**Keywords:** work–life balance, work–family balance, work–health balance, diversity in the workplace, job satisfaction, multiple regression, interaction analysis

## Abstract

To date, research directed at the work–life balance (WLB) has focused mainly on the work and family domains. However, the current labor force is heterogeneous, and workers may also value other nonworking domains besides the family. The aim of this study was to investigate the importance of other nonworking domains in the WLB with a particular focus on health. Moreover, the importance of the effects of the work–family balance (WFB) and the work–health balance (WHB) on job satisfaction was investigated. Finally, we explored how the effects of the WFB and the WHB on job satisfaction change according to worker characteristics (age, gender, parental status, and work ability). This study involved 318 workers who completed an online questionnaire. The importance of the nonworking domains was compared with a *t*-test. The effect of the WFB and the WHB on job satisfaction was investigated with multiple and moderated regression analyses. The results show that workers considered health as important as family in the WLB. The WHB explained more of the variance in job satisfaction than the WFB. Age, gender and parental status moderated the effect of the WFB on job satisfaction, and work ability moderated the effect of the WHB on job satisfaction. This study highlights the importance of the health domain in the WLB and stresses that it is crucial to consider the specificity of different groups of workers when considering the WLB.

## 1. Introduction

The term work–life balance (WLB) has gained increasing popularity in the public discourse [1]. It is a term that is commonly used in companies, especially large ones, and it is often said to be at the core of their corporate welfare, e.g., [2,3,4]. However, academic knowledge around the WLB concept is not as solid and extensive as the widespread use of the term would suggest [1]. Researchers have argued that WLB theoretical development has not kept pace with the popularity of the concept [5,6]. Among the many issues raised by recent critical reviews [1,7,8,9], the present study focuses on the limited consideration that has been given to the heterogeneity of the contemporary labor force in the WLB literature [8]. The extant research has largely assumed that the WLB is a concern mainly for working parents, where caring for dependent children is the relevant load in the life part of the WLB [1]. This became clear when we realized that the majority of the studies about the WLB actually only considered the family in the “life” part of the balance; that is, they considered the work–family balance (WFB) [6,9,10,11].

Currently, in addition to the massive presence of women in the labor market, which has fostered the literature about work–family balance, there is a highly increasing rate of active elderly workers, workers with a long-standing health problem or disability (LSHPD), single workers, and childless couples [12,13,14]. These workers have different needs and interests outside work. This situation places renewed importance on a key feature of the WLB: The importance that is attached to the many different life role changes from person to person [9]. Therefore, even if the family role remains central in nonworking life, it is important to recognize the value of other roles when conceptualizing and measuring the WLB [11]. The family may not be the most important part of the WLB in determining the positive outcomes of, for example, workers with chronic diseases for whom the management of health has great influence. From this perspective, Gragnano et al. [15] recently developed the concept and measure of the work–health balance (WHB), which is particularly relevant for elderly workers and workers with a LSHPD.

This study aims to contribute to the WLB research by comparing the relevance of other nonwork domains beyond family and considering the heterogeneity of the current labor force in studying the WLB. Specifically, we (a) investigate the perceived importance of other nonwork life domains beyond family, with a focus on health; (b) compare the influence of the WFB and the WHB on job satisfaction; and (c) examine how the effects of the WFB and the WHB on job satisfaction change according to different worker characteristics.

In subsequent sections of this article, we discuss the relationship between the WLB and the WFB, also considering different worker characteristics. We then introduce the concept of the WHB.

## 2. Theoretical Background and Hypotheses

### 2.1. Specific Nonwork Life Domains: Family and Health

The field of study about the work–life balance has had difficulty in establishing a commonly agreed-upon definition of the WLB [16]. A plethora of different conceptualizations exist in the literature, and many researchers have tried to summarize them [9,10,16]. After a review of the conceptualizations of the WLB in the literature, Kalliath and Brough [16] proposed a definition of the WLB that we endorse. “Work–life balance is the individual perception that work and nonwork activities are compatible and promote growth in accordance with an individual’s current life priorities” (p.326). A recent review indicated that a better work–life balance fosters not only job satisfaction, job performance, and organizational commitment but also life and family satisfaction [10]. The work–life balance also reduces stress-related outcomes such as psychological distress, emotional exhaustion, anxiety, and depression [10].

Research on work and nonwork interactions dates back to the mid-twentieth century, e.g., [17], and the issue has gained increasing importance in the popular press since the 1990s [16]. Today, there exists an extensive and growing body of research about the work–life balance [18], and the topic is of even more concern than in the past considering the new flexible ways of managing work (e.g., agile working, smart working, activity-based working, and flexible working).

Researchers have highlighted that the field of research about the work–life balance is itself “unbalanced.” The majority of studies on the work–life balance have focused only on work and family roles, that is, on the work–family balance [6,9,10,11]. For example, Casper et al. [9] reviewed the conceptual definition of the balance in the academic literature and found that 66% of the definitions focused only on work and family. In their review, Chang et al. [7] found that the WLB was studied specifically, not in the form of the WFB, in only 9% and 26% of the quantitative and qualitative studies reviewed, respectively. As a result, the knowledge acquired over time about the predictors and consequences of the balance with work is based mainly on the work–family balance [1,10].

Different types of the work–family balance have been studied in the literature. A general classification distinguished four types of influence that can occur between work and family based on their direction and valence [19]. When the effect is negative from the family domain to the work domain, it is called the family-to-work conflict. When the effect is still negative but from the work domain to the family domain, it is called work-to-family conflict. When the effect is positive, it is called enrichment and can have the same two directions; therefore, there is family-to-work enrichment and work-to-family enrichment. The work–family balance has been extensively studied in its negative form, work–family conflict [19]. However, since the 2000s, the scientific community has begun to focus on its positive form, work–family enrichment [20]. 

Many researchers have called for a real expansion of the WLB concept, such that the second arm of the balance—life—is not confined to the family role [6,9,19]. The call for an expansion of the concept is not only theoretically grounded but also related to recent changes in the labor market. The identification of the WFB as an indicator of the WLB was relatively effective and useful in recent decades, when the greatest change in workplace demographics was the increase in the participation of women, and the management of family and work roles for working women and dual-earner couples, especially those with children, became a central issue within organizations. Currently, workplace demographics are more heterogeneous. In addition to the massive presence of woman in the labor market, we are also seeing an increase in the rate of active elderly workers, workers with an LSHPD, single workers, and childless couples [12,13,14]. It is clear that an exclusive focus on family has become at least reductive when considering the WLB [1,21].

The majority of the studies that have investigated the work–nonwork balance without an exclusive focus on the family domain have considered nonwork to be unspecific, i.e., they have considered nonworking life in general, including nonfamily and family domains [19]. However, the consideration of the specific nonwork domains is essential to a full comprehension of the dynamics that influence the work–life balance in the heterogeneous working population, that is, the different, specific nonwork domains will have different levels of importance and different effects in the determination of the work–life balance among workers with diverse characteristics and needs outside work [11]. 

Based on the quality of life literature [22], the multiple identity perspective [23], and Super’s [24] life-space theory of career development, Keeney et al. [11] identified eight nonwork domains of relevance in the WLB: education, health, leisure, friendships, romantic relationships, family, household management, and community involvement. The importance that individuals give to the different domains varies from person to person [11]. Moreover, the relative importance of these life domains is likely to change over time within the same person because of changes in interests and life circumstances [24]. Thus, it is crucial to understand whether the other nonwork domains are as important as family and under which circumstances the priorities change. Among the domains that were detected by Keeney et al. [11], there was health. This is relevant because to our knowledge, for the first time in work–life balance literature, it has been recognized that health management can conflict with work activity.

As stated, the relevance of health to the work life derives from an increase in the rate of workers with an LSHPD and elderly workers, both with a higher incidence of health problems. In 2017, 27.8% of the European Union (EU) workers reported an LSHPD, and 19% of the employed persons in the EU were 55 years of age or older [25,26]. There is, however, another reason that makes the health domain relevant even for “healthy” workers. A paradigm shift has occurred in the planning and delivery of healthcare. People are now expected to actively manage their healthcare. Theorizations in the field of public health and in medicine have indicated that it is strategic for healthcare systems to have informed patients who are more directly responsible for their health and care management [27]. This has been paired with an increasing focus on health promotion that is based, partly but strongly, on good individual healthy behaviors [28]. Therefore, the workers, not just the sick ones, must take on a somewhat active role in the health domain of life, which may be more or less compatible with the working role.

In light of this literature and considering the life domains defined by Keeney et al. [11], we hypothesized that family is still central in the WLB of workers but that the health domain also has an equally important role. Therefore, if the workers were asked directly:
**Hypothesis** **1** **(H1):***Workers will indicate that the family and health domains are more important than the other life domains in the WLB process*.

### 2.2. Consequences of Work–Family Balance: Job Satisfaction

Many studies have analyzed individual consequences of the different types of the work–life balance, and several meta-analyses have summarized the literature about the correlates of work–family conflict [29,30,31,32] and work–family enrichment [20]. Work–family conflict, in both directions, has been consistently found to be associated with work-related, family-related, and domain-unspecific outcomes. Specifically, among the many outcomes that are associated with work–family conflict in a statistically significant manner, the ones that were more strongly associated were organizational citizenship behavior, work-related and general stress, burnout and exhaustion, and job, marital and life satisfaction [29]. Far fewer studies exist for work–family enrichment, but by comparing the two extant bodies of literature, it is possible to note that the effect sizes of work–family enrichment are comparable to those of work–family conflict [20,29]. For simplicity and because more studies are needed about the relationship between work–family enrichment and conflict [33], which goes beyond the objectives of this research, we considered only the conflict, in both directions, in our study.

Among the literature considering work-related outcomes, job satisfaction has been the most studied variable [29]. Job satisfaction represents the extent to which workers like or dislike their job [34]. Job satisfaction is a central variable in organizational behavior research. Spector [34] ascribed its importance to three main reasons. Job satisfaction is an indicator of well-being and psychological health, it is related to many behaviors of the worker that are positive for the organization, and finally, it is a very useful indicator of organizational problems when its level is low. In fact, job satisfaction is highly related to burn-out, self-esteem, depression, anxiety and, to a lower extent, perceived physical illness [35]. It is consistently correlated with job performance [36] and with four dispositional traits predictive of job performance: self-esteem, generalized self-efficacy, locus of control, and emotional stability [37]. Job satisfaction has also been found to be a significant predictor of turnover and turnover intention [38,39]. 

Job satisfaction is also related to the work–family balance. The meta-analysis conducted by Amstad et al. [29] reported that the correlation with job satisfaction was stronger for work-to-family conflict (weighted mean correlation = −0.26) than for family-to-work conflict (weighted mean correlation = −0.13). Theoretically, the work–family balance affects job satisfaction because an incompatibility between two personally relevant roles creates negative states and feelings. Following the principle that when something threatens something else personally relevant, the first is appraised negatively with negative emotion [40], and a role that interferes with the fulfilment of another personally relevant role is negatively evaluated. Specifically, a negative evaluation of an individual’s job is formed (i.e., low job satisfaction) depending on the extent to which the job threatens the fulfillment of the family role [41]. This explanation justifies why family-to-work conflict has been found to have a lower correlation with job satisfaction than work-to-family conflict. In fact, provided that both conflict directions may generate a strain in both domains, the family-to-work conflict will generate a low family satisfaction—instead of a low job satisfaction—because the family role interferes with the work role, and the negative evaluation will be toward the source of the interference [41]. This was supported by the meta-analysis conducted by Amstad et al. [29], who found that work-to-family conflict was more strongly correlated with work-related outcomes than family-related ones and that the opposite was true for family-to-work conflict. Based on these premises, we hypothesized that:
**Hypothesis** **2a** **(H2a):**Work-to-family and family-to-work conflict will be significantly and negatively related to job satisfaction.
**Hypothesis** **2b** **(H2b):**The relationship between work-to-family conflict and job satisfaction will be greater than the relationship between family-to-work conflict and job satisfaction.

### 2.3. Consequences of Work–Health Balance

The present study aimed to expand the knowledge about the nonwork life domain other than family, specifically the health domain. Despite the importance of the life domain of health, the literature has not offered many studies that consider health in the WLB process or measurement instruments that are specifically designed for the purpose [42]. Considering the literature about job retention and the quality of working life among workers with an LSHPD [43,44], Gragnano et al. [15] conceptualized the work–health balance (WHB) as a state in which the worker feels able to effectively balance health and work needs, arising from the perception of how much the characteristics of one’s work are a barrier to health needs and counterbalanced by the evaluation of the helpfulness of the working environment to meet health needs.

Health needs are understood here in a broad sense, covering not only the care needs of workers with chronic illnesses or conditions but all the needs that a worker considers necessary to adequately care for his or her health. From the definition, a measure of the WHB has been developed. The WHB questionnaire measures three distinct constructs: work–health incompatibility, health climate and external support [15]. The first construct measures how much work commitments hamper the desired management of health. The last two constructs measure the helpfulness of the working environment for health needs. The health climate detects the extent to which workers perceive that management is truly interested in their employees’ health, whereas external support identifies the perception of the level of help available for health problems in the workplace in the form of support from the supervisor and work flexibility.

Studies have shown that elderly workers and workers with an LSHPD have more difficulties in reaching a good WHB [45,46]. In addition, it has been shown that among workers who stop working for cardiovascular diseases, the process of returning to work is faster for those who have a good WHB [47]. With low levels of the WHB, the rates of presenteeism, emotional exhaustion, workaholism and general psychological distress (GHQ) increase [15,48]. In contrast, a good WHB is associated with greater work autonomy, job engagement, and job satisfaction [15,49,50,51].

In the WHB, a good balance generates job satisfaction because the work role is not a threat to the management of health. A low level of work-to-family conflict generates job satisfaction because the work role is not a threat to the family domain. Because the two domains at risk are different, the proportion of the job satisfaction variance that is explained by the WHB is expected to not overlap, to a great extent, with the proportion that is explained by the work-to-family conflict. Moreover, in the current working context, characterized by a great heterogeneity of the contemporary labor force with a substantial proportion of elderly workers and workers with an LSHPD, as well as with the increasing spread of a health care system that is based on the active and informed role of patients, we expect the WHB to be as important as work-to-family conflict in shaping attitudes toward job and job satisfaction. Therefore, we hypothesized that:
**Hypothesis** **3a** **(H3a):**The WHB will have a significant positive effect on job satisfaction.
**Hypothesis** **3b** **(H3b):**The effect size of the WHB on job satisfaction will be at least as large as that of work-to-family conflict.

### 2.4. The Heterogeneity of the Labor Force and WLB

As stated before, the present study focuses on the problem of the limited consideration that has been given in the WLB literature to the heterogeneity of the contemporary labor force [8]. The current labor force is characterized not only by a greater female presence but also by an increasing rate of elderly workers, workers with an LSHPD, single workers, and childless couples [12,13,14], all with different needs and with a different levels of importance that are given to their various nonworking roles [1].

This last consideration is particularly relevant in the context of the WLB because the balance is not absolute; rather, it depends on the importance that is given by the worker to the various roles. Therefore, when studying the effect of the WLB on outcomes by using concepts and measures such as work-to-family conflict or the WHB, which measure the balance between a specific nonwork role and work, it is theoretically appropriate to expect that the studied effect will vary based on the importance that is given by the worker to the nonwork role under consideration. In other words, the perception of an imbalance between a specific nonworking role and work will have a negative effect on the outcome to the extent that the nonworking role in question is important for the worker.

Despite the centrality of individual priorities in the definition of the WLB [9,10,16], surprisingly few studies have explored how individual priorities moderate the effect of the WLB on outcomes [6,29,52], which is a symptom of the limited consideration of diversity in the labor force by the WLB literature [1,8]. Individual differences have been considered as predictors of differences in the level of balance [10,53] instead of as moderators of the effects of the balance on the outcomes. Crooker et al. [21] developed a theoretical framework that extensively considered differences in individual value systems as moderators, but this study was focused on the genesis of the WLB instead of its consequences. 

In the present study, we considered four variables (i.e., age, gender, parental status, and work ability) that, according to the literature, moderate the relationship between the WFB and job satisfaction or, alternatively, the relationship between the WHB and job satisfaction. The hypothesis is that individual conditions and characteristics that increase (or decrease) the importance that is given by the worker to the family or health domain will increase (or decrease) the effect that the work–family balance or the WHB has on job satisfaction.

Gender has been studied in the WLB literature as a possible predictor of different levels of the work–family balance. The hypothesis has been that, since family responsibilities usually pertain more to women, women have worse levels of the work–family balance, but these studies have not consistently supported this hypothesis [54]. However, research has still indicated that there are significant disparities between men and women pertaining to the work–family balance [55]. There have also been studies that have indicated that women do value family more than men, and the opposite has been shown to be true for work [56,57]. This is consistent with other studies that have indicated a stronger effect of the work–family balance on job satisfaction [58,59] and negative emotional responses [60] for women. Based on these premises, we hypothesized that:
**Hypothesis** **4a** **(H4a):**The negative effect of work–family conflict (work-to-family and family-to work) on job satisfaction will be stronger for women than for men.

Similarly, there is evidence that parents experience more problems with the work–family balance than workers without children (for a meta-analysis, see [61]). This is often because family-related demands are higher for parents [62]. However, we also sustain that the importance that is given to the family domain is higher for workers with children than for those without. Thus, we hypothesized that:
**Hypothesis** **4b** **(H4b):**The negative effect of work–family conflict (work-to-family and family-to work) on job satisfaction will be stronger for workers with children than for those without.

Socioemotional selectivity theory (SST) [63] sustains that individuals have an intrinsic perception of the time left in their life—the future time perspective—and based on that, they adjust their preferences and behavior. A shortened future time perspective promotes the pursuit of short-term emotion-related goals, such as positive emotional and psychological well-being, and it devaluates long-term goals, such as the development of skills or career advancements [63]. In the WLB literature, SST implies that elderly workers, who have a shorter future time perspective, should consider family relationships more important than work [64]. Therefore, a high level of work-to-family conflict will affect elderly workers and their evaluation of job satisfaction more than younger worker. In line with this, Treadway et al. [65] found that, in the presence of a high work-to-family conflict, workers with a more constrained future time perspective experienced a lower continuance commitment than employees with a less shallow future time perspective.

**Hypothesis** **4c** **(H4c):**
*The negative effect of work–family conflict (work-to-family and family-to-work) on job satisfaction will be stronger for elderly workers than for younger workers.*


Because increasing age is associated with higher morbidity, (multiple) chronic conditions, and higher use of health services [66], the importance of the health domain is expected to be higher among elderly workers than younger workers. Therefore, we hypothesized that:
**Hypothesis** **4d** **(H4d):**The positive effect of the WHB on job satisfaction will be stronger for elderly workers than for younger workers.

Finally, work ability is expected to play a role in association with the WHB. Work ability represents the perceived ability to do one’s job effectively and to continue to do so in the near future when considering personal health problems and resources [66]. Thus, in the life of workers with a low work ability, the health domain generally has more importance than workers with a high work ability because health is a current problem. Considering this, we hypothesized that:
**Hypothesis** **4e** **(H4e):**The positive effect of the WHB on job satisfaction will be stronger for workers with a low work ability than for those with a high work ability.

## 3. Materials and Methods 

### 3.1. Sample and Procedure

The study involved workers of full age under an employment contract. Entrepreneurs and self-employed workers were excluded. We distributed the link to the online questionnaire with a brief description of the research through social networks (i.e., Facebook and LinkedIn), messaging applications, and email. To begin the assessment, the participants had to read and approve an informed consent form to freely decide whether to participate in the research. The informed consent provided informed about the aim of the study and the procedures to collect the data, and it ensured that there were no potential risks or costs involved. The research team assured the anonymity and confidentiality of the participants’ responses throughout the entire study process. The contact details of the researcher in charge were provided in the event of any further questions. The study was conducted in accordance with the ethical standards set by the Declaration of Helsinki and was approved by the Ethical Committee of the University of Milano-Bicocca (Prot.160-2014). The number of subjects that started the questionnaire was 350. However, the dataset used in the analyses contained 318 responses after excluding 32 questionnaires because they were substantially incomplete; that is, the subjects opened the online page of the questionnaire but did not answer any questions. These values represent a completion rate of 91%. All participants lived in Italy; 90% lived in northern Italy. Overall, 37%, 28%, and 35% of the respondents were between 20 and 30, 31 and 44, and 45 and 60 years old, respectively. The proportion of men and women, as well as people with and without children, was balanced in the sample (56% women and 58% with children). Among the 134 workers with children, 49%, 43%, and 8% of the respondents had one, two, and three or more children, respectively. The workers with one or more children under the age of twelve were 51%. Most of the respondents had a partner (76%) and at least an upper secondary school diploma (93%). Most of the participants worked full-time (85%) with an open-ended contract (79%) as a white-collar worker (72%). Table 1 presents detailed descriptive statistics of the sample. 

### 3.2. Measures

The sociodemographic information described above was provided by the respondents at the beginning of the online questionnaire.

Based on the instrument developed by Keeney et al. [11] to evaluate the importance in the WLB attached to the different life domains (family, health, household management, friendship, training activities, favorite leisure activities, and community involvement), respondents were asked “How important is it in your life to reconcile work with *…?*”. The question was asked, changing the final part, for all of the seven domains of life considered. The response scale was a 10-point scale from 1 (not at all important) to 10 (extremely important).

Two forms of the WLB were measured: the work–family balance and the work–health balance. The work–family balance was measured in the form of the work-to-family conflict (WFC—three items, α = 0.79) and family-to-work conflict (FWC—three items, α = 0.72) with the abbreviated version of the measure of work–family conflict [67]. Answers were given with a five-point Likert scale, from 1 (completely disagree) to 5 (completely agree). The work–health balance was measured with the Work–Health Balance Questionnaire [15], which was composed of three subscales: work–health incompatibility (WH—six items, α = 0.84), health climate (HC—five items, α = 0.92), and external support (ES—six items, α = 0.81). The total WHB score was calculated by subtracting WHI from the mean of HC and ES. Answers were given according to a five-point rating scale from 1 (completely disagree) to 5 (completely agree) for WHI and from 1 (never) to 5 (always) for HC and ES.

Work ability, the perceived ability to do one’s job effectively and to continue to do so in the near future when considering personal health problems and resources, was measured with the Work Ability Index (WAI) [68]. The index was calculated from seven factors (α = 0.79) for a total of 10 items with different rating scales.

Job satisfaction was measured with a single item that asked respondents to rate their overall satisfaction with their job on a 5-point scale from 1 (not at all satisfied) to 5 (fully satisfied). The reliability and validity of the single-item measure to assess job satisfaction has been established [69].

Harman’s single-factor test was adopted to check for a common method bias. The first factor explained 27% of the variance. Given that this fell below the threshold of 50%, the common method bias does not appear to have been a significant factor in this study. The results of the explorative factor analysis performed for the Harman single-factor test are available in the online Appendix A of this article.

### 3.3. Data Analysis

All data analyses were performed by using R [70]. The different life domains were ordered according to the mean importance to the WLB that was attached to them by the respondents. Mean and standard deviations were provided for all the life domains. To test whether family and health domains were considered more important than the other life domains in the WLB (H1), the mean of the importance that was attached to health and family were compared to the mean of the importance that was attached to all the other life domains with a paired t-test. Even if no hypothesis was formulated specifically on this point, we explored whether the family and health domain were considered equally important. A paired t-test between the importance ascribed to family and to health was performed.

The hypotheses about the direction and effect size of work-to-family conflict, family-to-work conflict, and the WHB on job satisfaction (H2a,b and H3a,b) were tested with a multiple linear regression with job satisfaction as the dependent variable and work-to-family conflict, family-to-work conflict and the WHB as independent variables. To evaluate the relative importance of these predictors to the multiple regression model just described, we used the Lindeman, Merenda, and Gold’s metric (LMG) and reported the standardized *β*. The LMG expresses the squared semipartial correlation that was averaged across all possible ordering of the predictors. Since each order of predictors yields a different decomposition of the model sum of squares, the variance of the dependent variable that is explained by a predictor in a multiple regression varies according to the sequential order in which a predictor is entered into the model in relation to the other predictors. LMG averaged this value for all the possible orders of entry [71]. As a result, LMG considers both the predictor’s direct effect and its effect when combined with other predictors. Conversely, the standardized *β* represents only the incremental contribution of each predictor when combined with all remaining predictors [71,72].

This model, as well as the other following models, was controlled for age, marital status, and parental status. The control variables to be included were chosen with a backward model selection by the Akaike information criterion (AIC) from an initial model that included age, gender, education level, marital and parental status, job role, type of contract, and working hours. These preliminary analyses are available in the online Appendix A of this article.

Finally, the hypotheses about the moderation of the relation between the work–family balance and/or the WHB with job satisfaction (H4a–e) by individual characteristics (age, gender, parental status, and work ability) were tested with several models—one per moderator—with interaction effects. Continuous variables involved in the interaction were centered on the mean.

## 4. Results

### 4.1. Perceived Importance of Family and Health Domain

The mean and standard deviation of the importance that is attached to the different life domains, ordered by their importance, are listed in Figure 1.

The first paired t-test resulted in a significant difference in the mean of the importance that was attached to health and family (M = 9.27 and SD = 1.04) and those ascribed to the other life domains (M = 7.3 and SD = 1.32); *t*(317) = 25.7 and *p* < 0.001. This result supported H1a, that is, the health and family domains were considered to be more important than the other domains in the WLB.

Moreover, the second paired t-test resulted in a nonsignificant difference in the importance that is attached to health (M = 9.29 and SD = 1.18) and those attached to family (M = 9.25 and SD = 1.3); *t*(317) = 0.57 and *p* = 0.57. This exploratory analysis showed that health and family are life domains considered of equivalent importance in the WLB.

### 4.2. Consequences of Work–Family and Work–Health Balance on Job Satisfaction

Table 2 presents the result of the first model that tested the effects of work-to-family conflict, family-to-work conflict, and the WHB on job satisfaction (R^2^ = 0.28, F_(6/308)_ = 20.24, and *p* < 0.001). 

The model resulted in a significant negative effect of work-to-family conflict and a nonsignificant effect of family-to-work conflict, thus partially supporting H2a. The LMG of work-to-family conflict on job satisfaction (LMG = 0.08) was eight times greater than that of family-to-work conflict (LMG = 0.01). Moreover, the former was statistically significant, while the other was not. These results fully supported H2b. Considering the effect of the WHB on job satisfaction, the model estimated a significant positive effect, supporting H3a. Moreover, the variance that was explained by the WHB (LMG = 0.16) was twice as much as the variance that was explained by work-to-family conflict (LMG = 0.08), supporting H3b.

### 4.3. Moderators of the Effects of Work–Family and Work–Health Balance

Table 3 reports models 1 and 2, which tested the moderating effect of gender and parental status, respectively.

Model 1 (R^2^ = 0.30, F_(9/305)_ = 14.54, and *p* < 0.001) in Table 3 showed a significant negative interaction of gender with work-to-family conflict but no interaction with family-to-work conflict. The interaction indicates that the negative effect of work-to-family conflict on job satisfaction was stronger among women than among men. To facilitate the interpretation, the interaction effect is depicted in Figure 2. This result partially supported H4a: The effect of the work–family balance, specifically of work-to-family conflict, on job satisfaction was stronger among women than among men.

Model 2 (R^2^ = 0.31, F_(8/306)_ = 17.46, and *p* < 0.001) in Table 3 again showed a significant interaction of work-to-family conflict with the moderator (i.e., parental support) but no interaction of the moderator with family-to-work conflict. The interaction indicates that the negative effect of work-to-family conflict on job satisfaction was stronger among workers with children than among those without. This interaction effect is depicted in Figure 3. This result partially supported H4b: The effect of the work–family balance on job satisfaction, specifically of work-to-family conflict, is stronger among workers with children than among those without.

Table 4 reports models 3 and 4, which tested the effects of two moderators—age and work ability, respectively. Model 3 (R^2^ = 0.31, F_(9/305)_ = 15.28, and *p* < 0.001) in Table 4 showed a significant interaction of age with work-to-family and family-to-work conflict but no interaction with the WHB.

The interactions showed that the negative effect of work-to-family conflict on job satisfaction increased with age (Figure 4a), whereas family-to-work conflict appeared to have a positive effect for older workers (Figure 4b). These results again supported H4c only for work-to-family conflict, whereas they showed an unexpected positive effect of family-to-work conflict on job satisfaction among the elderly. In contrast, the results did not support H4d because the effect of the WHB on job satisfaction did not seem to increase with age.

Model 4 (R^2^ = 0.33, F_(8/306)_ = 18.18, and *p* < 0.001) in Table 4 showed a significant negative interaction between the WHB and work ability. The interaction showed that the positive effect of the WHB on job satisfaction decreased with the increase in work ability (Figure 5). This result supported H4e: The positive effect of the WHB on job satisfaction increased with the decline in work ability.

## 5. Discussion

This study aimed to verify the importance of different, specific nonwork domains in the work–life balance process, with a focus on family and health. We also investigated the impact of the work–family balance (in both directions) and the work–health balance on job satisfaction and how the heterogeneity of the current workforce modifies these relationships.

The results supported the first hypothesis. As hypothesized, when considering their work–life balance, the workers attached more importance to the health and family domains than to the other nonwork domains. A further analysis showed that the health and family domains were given similar importance. This result was the starting point of the entire study and justified the inclusion of the concept of the work–health balance. The workers rated family and health as 25% more important than the other nonwork life domain in their work–life balance. The fact that health was important as family is a relevant result, and it was found to be even more important when we analyzed the sample. Indeed, there were no apparent sample characteristics that made this sample more exposed to health issues than the general population. This fact suggests that researchers and companies should pay more attention to the health domain even for workers that are not affected by severe or chronic health conditions.

The second hypothesis concerned the effect of the work–family balance on job satisfaction. The work–family balance was supposed to affect job satisfaction, and work-to-family conflict was supposed to be more important than family-to-work conflict. The results supported this hypothesis and, consistent with other studies, the effect of family-to-work conflict on job satisfaction was smaller than that of work-to-family conflict and was statistically not significant [73]. This result can be explained in light of the appraisal theory [40]: If work threatens family life (work–life conflict), work will be negatively appraised; if family issues threaten work participation (family–work conflict), family, not the work, will be negatively appraised [41]. Consistent with a prior meta-analysis [29], these results support the “matching-hypothesis” (work-to-family conflict affects the work domain more, whereas family-to-work conflict affects the family domain more) as opposed to the “cross-domain hypothesis” (work-to-family conflict affects the family domain more, whereas family-to-work conflict affects the work domain more). Our study provides new evidence in this sense because the regression model was controlled for the work–health balance and because of the adoption of the LMG metric.

The third hypothesis investigated the effect of the WHB on job satisfaction and its importance relative to work–family conflict. As hypothesized, the WHB had a positive and statistically significant relationship with job satisfaction, and its importance was two times greater than that of work-to-family conflict. This result supports the usefulness of the specific instrument, the WHB questionnaire, and confirms the importance of filling the gap in the literature [42] by introducing the health domain into the concept of the work–life balance. Even if our results cannot be considered definitive in saying that the health domain is more important than the family domain in the genesis of job satisfaction, they clearly indicate that, when investigating or promoting work–life balance, considering the WHB is at least as important as considering the work–family balance. The common practice of considering the work–life balance as an issue that is related only to family is wrong and limits the possibility to explain work phenomena through the lens of the work–life balance.

The fourth hypothesis was related to the moderation of the effects of work–family conflict and the WHB by specific work characteristics (i.e., gender, parental status, age, and work ability) on job satisfaction. All three hypothesized moderators of the effect of work-to-family conflict on job satisfaction (i.e., gender, parental status, and age) were supported, whereas only one moderator of the family-to-work conflict effect (i.e., age) was sustained. Of the two hypothesized moderators (age and work ability) of the WHB effect on job satisfaction, only the interaction with work ability was supported. 

In particular, the impact of work-to-family conflict on job satisfaction was greater for women (H4a), parent workers (H4b), and elderly workers (H4c). The reason for this moderation effect is likely due to the difference in salience of the family domain attached by the groups of workers. Women are likely to evaluate family as more central in their lives than men because of widespread cultural norms and gender-differentiated values [56,57]. Likewise, parents are likely to give more salience to family than people with no children because of cultural norms and, possibly, because of a “self-selection process” that brings people with a high salience of family to be more prone to parenthood than people with a low salience [62,74]. Given such result, it is possible, and should be tested in future studies, that being responsible for eldercare, beyond generally increasing the level of work-to-family conflict, also increases the impact of work-to-family conflict on job satisfaction. Finally, as implied by the socioemotional selectivity theory, elderly workers are likely to consider family relationships more important than younger workers because of a shorter future time perspective [64]. 

Given the theoretically coherent nonsignificant main effect of family-to-work conflict on job satisfaction, it was not surprising that the hypothesized moderators of its effects were not relevant. However, the moderation of the effect of family-to-work conflict on job satisfaction by age was significant and indicated that among older workers, a higher level of family-to-work conflict was related to higher job satisfaction. A further analyses showed that the effect of family-to-work conflict was nonsignificant among workers of 27 (the first quartile) and 38 years of age (the mean age), but this effect was statistically significant among workers of 49 years of age (the third quartile). The interpretation of this effect is hazardous with the data at hand. Further studies should investigate this effect while also considering the cross-sectional nature of our study. In fact, it is not possible to exclude that the found relationship was inverse. That is, older workers with higher job satisfaction perceived a higher family-to-work conflict because of a greater importance that is attached to the work domain than other elderly workers with lower job satisfaction.

Concerning the WHB, we hypothesized that its effect on job satisfaction was stronger among older workers (H4d) and workers with a lower WAI score (H4e). Since the interaction term was not significant in the case of age, H4d was not supported. Our results showed that a good WHB was associated with an equally high job satisfaction among all ages. We believe this is simply because, in our sample, the importance that was given to the health domain was not associated with age. This idea was supported by post hoc analyses that correlated the importance that was given by the workers to the health domain with their age, which resulted in a nonsignificant correlation (*r* = −0.09, *t* = −1.62, and *p =* 0.11). We believe this result indicates that the health domain is crucial for both younger and older workers. There is the possibility that the WHB is a very important dimension at all ages—not only for elderly workers as originally intended [15]. In contrast, our results supported H4e. With the decline of the WAI, that is, with more health problems affecting job activity, the importance of the positive effect of the WHB on job satisfaction was increased. As proposed elsewhere [15], workers who are more vulnerable to health problems had a greater gain from their work situation with a good balance between health needs and work demands than healthy workers.

Overall, the results regarding the hypothesized moderators indicate that it is crucial to take into account the heterogeneity of the current workforce and to consider the specificity of different groups of workers when considering the WLB. From the outset, most definitions of the work–life balance have stressed the fact that it is not possible to identify an absolute optimal balance because it depends on the importance that the worker gives to the different domains of life [1,9,10,16]. Despite being theoretically clear, individual differences have been mainly overlooked in the WLB literature. Our study presents strong evidence that the issue must be considered, especially in light of the large presence of women, elderly individuals, people with an LSHPD, singles, and childless couples in the labor force [12,13,14].

The current study presents some limitations to consider when interpreting the results. First, the study design was cross-sectional. This limits our confidence in determining the cause and effect in the relationships between the considered variables. We based our considerations on a strong theoretical basis [10,29], but longitudinal studies are needed to replicate our findings.

Second, we adopted an online recruitment procedure that has the problem of a participant selection bias because of the self-selection of participants [75]. The online recruitment made our sample not representative of the entire working population, but this was beyond our intent. As explained by Landers and Behrend [76], when the aim is to test theoretically relevant hypotheses, as in our study, sample representativeness is less crucial than when a study aims to estimate the presence and the level of one or more variables in the workers’ population. Of course, our results must be replicated in other samples to increase their generalizability. By comparing our sample characteristics with data representative of the employees in north Italy [77] (data shown in the online Appendix A of this article), it is possible to note some differences in the proportions of job roles, type of occupations, and levels of education that are worth being mentioned. Specifically, like many studies in the WLB literature [7], in our sample, there was an over-representation of white-collar workers and an under-representation of blue-collar workers. There was an over-representation of clerical support workers and an under-representation of factory workers, skilled laborers, building workers, elementary occupations, and services and sale workers. Finally, the level of education of the sample was higher than in the general population of employees in north Italy. Given these specificities, it will be necessary to test whether the same results hold across samples with an appropriate representation of factory workers, skilled laborers, building workers, elementary occupations, and services and sale workers, as well as employees with a lower level of education.

Third, the measure of the importance that workers gave to the different life domains was based on the instrument of Keeney et al. [11], but the final instrument was created ad hoc for this study. Therefore, the measurement instrument may have biased the results regarding the importance of the different life domains. However, it should be considered that the questions that were posed to the participants were quite straightforward, and the values obtained for each domain were plausible and not extreme. Even if the instrument was not fine-tuned for exact comparisons, we believe it was appropriate for the aim of the study. The cited limitations warn against an unconditional generalization of the results of this study that, instead, have to be replicated with stronger research designs and other samples of workers.

## 6. Conclusions

The health issue has emerged in the organizational literature as a central topic. It no longer pertains to only small groups of workers with severe health problems. The changes in the labor force and of the patient’s role in the health system have made it impossible to consider the management of health as an exclusively nonworking activity. This study shows that workers are aware of the importance of the health domain for achieving a good work–life balance. Our results indicate health as a fundamental domain in the work–life balance dynamic that is as important as the family domain, if not more so. Researchers and practitioners should therefore consider the health domain in addition to the family domain when investigating the work–life balance. 

By showing the differences in the effects of the work–family balance and the work–health balance on job satisfaction for different categories of workers, the present study demonstrates the importance of individual differences in the work–life balance process. That is, the balance between work and life is not absolute, but it is related to the importance that is given by the worker to the various domains. This relationship is of prominent importance in the current heterogeneous labor force.

Finally, our results provide evidence, to be replicated, that the importance of the work–health balance is not related to age, as previously believed; but only with the health condition.

Overall, this study is relevant for the work–life balance literature because, to the best of our knowledge, it is the first to consider the work–health balance. Moreover, it is one of the few studies that, through moderation analyses, investigates how the effect of the work–life balance on a relevant outcome changes according to workers’ characteristics.

## Figures and Tables

**Figure 1 ijerph-17-00907-f001:**
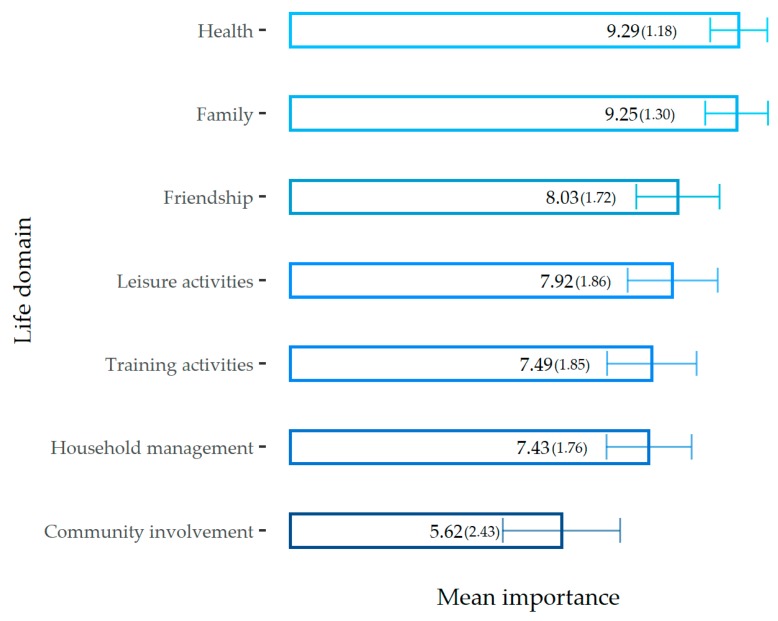
Mean importance and standard deviation of the seven life domains.

**Figure 2 ijerph-17-00907-f002:**
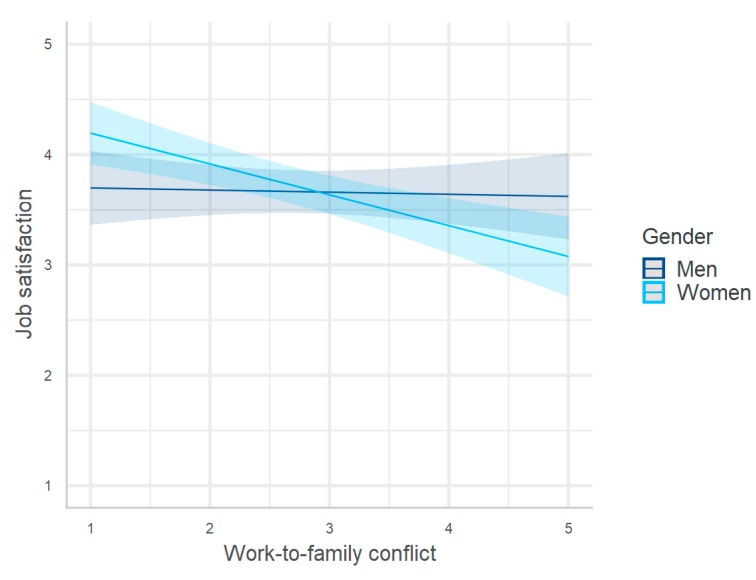
Moderating effect of gender on the relationship between work-to-family conflict and job satisfaction.

**Figure 3 ijerph-17-00907-f003:**
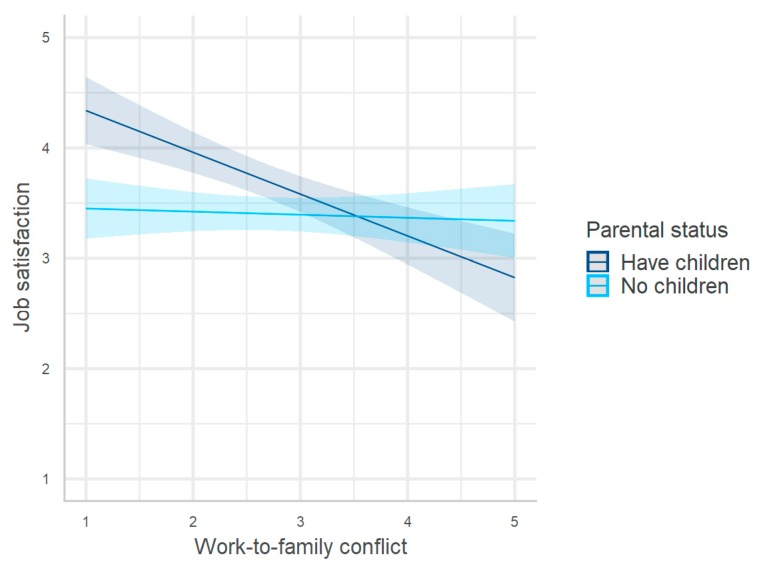
Moderating effect of parental status on the relationship between work-to-family conflict and job satisfaction.

**Figure 4 ijerph-17-00907-f004:**
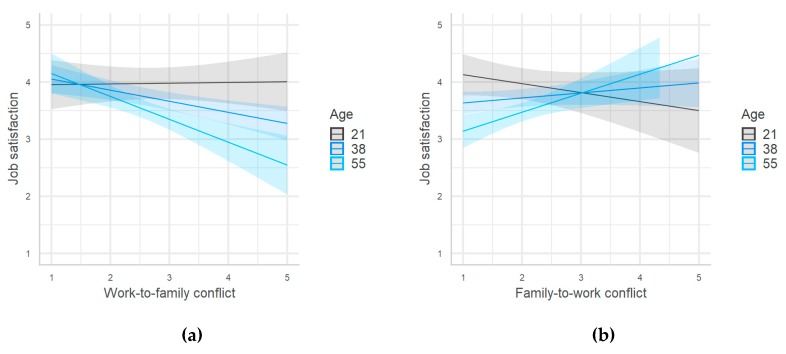
(**a**) Moderating effect of age on the relationship between work-to-family conflict and job satisfaction. (**b**) Moderating effect of age on the relationship between family-to-work conflict and job satisfaction.

**Figure 5 ijerph-17-00907-f005:**
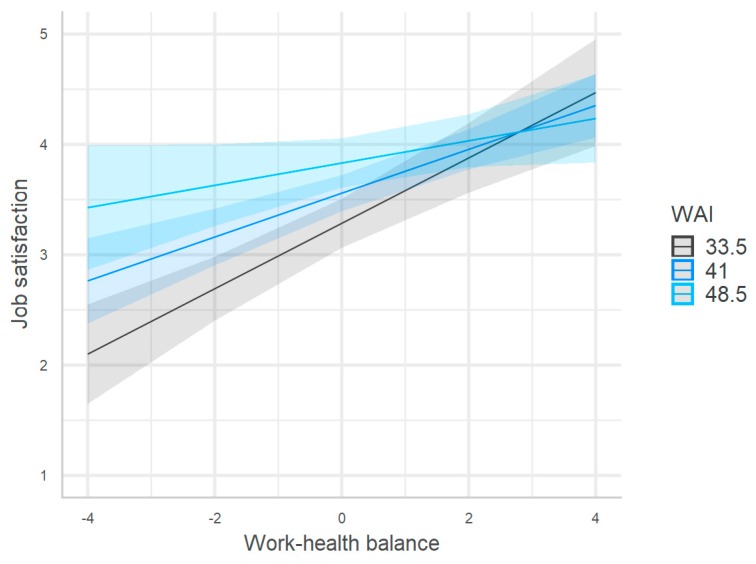
Moderating effect of work ability on the relationship between the work–health balance and job satisfaction.

**Table 1 ijerph-17-00907-t001:** Descriptive statistics of the sample (N = 318).

Variable		% (*n*)
Mean age (SD)		38.14 (11.59)
Female gender		56% (177)
Education level	Primary and lower secondary school	7% (22)
	Upper secondary school	53% (169)
	University or higher	40% (127)
Parental status	Have children	42% (134)
Marital status	Have a partner	76% (241)
Type of contract	Open-ended	79% (251)
	Fixed-term	17% (53)
	Other	4% (14)
Main work activities	Physical	7% (23)
	Intellectual	80% (254)
	Both	13% (41)
Job role	Manager	3% (9)
	Supervisor	13% (41)
	White collar	72% (228)
	Blue collar	8% (27)
	Other	4% (13)
Working hours	Part-time	15% (49)
	Full-time	85% (269)
Occupation	Chief Executives, Senior Officials and Legislators	1.3% (4)
	Professional	11.4% (36)
	Technicians and associate professionals	31.9% (101)
	Clerical support workers	45.4% (144)
	Service and sales workers	3.5% (11)
	Factory worker, skilled laborer, building workers	5.3% (17)
	Other occupations	1,2% (4)

**Table 2 ijerph-17-00907-t002:** Adjusted effects of work-to-family conflict, family-to-work conflict, and the work–health balance (WHB) on job satisfaction.

Variable	b(se^1^)	*t*	β	LMG
Work-to-family conflict	−0.17 (0.06)	−2.97 **	−0.18	0.08
Family-to-work conflict	0.07 (0.07)	1.03	0.05	0.01
Work–health balance	0.24 (0.04)	6.52 ***	0.39	0.16
Age	−0.01 (0.004)	−2.49 *	-	-
Marital status (no partner)	0.24 (0.12)	2.11 *	-	-
Parental status (no children)	−0.30 (0.12)	−2.57 *	-	-

*** = *p* < 0.001; ** = *p* < 0.01; * = *p* < 0.05; ^1^ se = standard error.

**Table 3 ijerph-17-00907-t003:** Adjusted effects of work-to-family conflict, family-to-work conflict, and the WHB on job satisfaction.

Variable	Model 1		Model 2	
	b(se)	*t*	b(se)	*t*
Work-to-family conflict (WFC)	−0.02(0.08)	−0.25	−0.38(0.08)	−4.71 ***
Family-to-work conflict (FWC)	0.03(0.09)	0.32	0.21(0.1)	2.23 *
Work–health balance (WHB)	0.25(0.04)	6.60 ***	0.25(0.04)	6.8 ***
Moderator ^1^	0.61(0.32)	1.91	−0.77(0.32)	−2.4 *
WFC *moderator ^1^	−0.26(0.10)	−2.69 **	0.35(0.10)	3.62 ***
FWC *moderator ^1^	0.08(0.13)	0.6	−0.24(0.13)	−1.88
Age	−0.01(0.005)	−2.46 *	−0.01(0.005)	−2.4 *
Marital status (no partner)	0.28(0.12)	2.43 *	0.26(0.11)	2.28 *
Parental status (no children)	−0.29(0.12)	−2.48 *	-	-

*** = *p* < 0.001; ** = *p* < 0.01; * = *p* < 0.05; ^1^ In model 1, the moderator is gender (female); in model 2, the moderator is parental status (no children). Continuous variables in the interactions have been centered on the mean.

**Table 4 ijerph-17-00907-t004:** Adjusted effects of work-to-family conflict, family-to-work conflict, and the WHB on job satisfaction.

Variable	Model 3		Model 4	
	b(se)	*t*	b(se)	*t*
Work-to-family conflict (WFC)	−0.20(0.06)	−3.48 ***	−0.15(0.06)	−2.75 **
Family-to-work conflict (FWC)	0.09(0.07)	1.37	0.07(0.06)	1.10
Work-health balance (WHB)	0.25(0.04)	6.7 ***	0.20(0.04)	5.11 ***
Age	−0.03(0.007)	−3.8 ***	−0.01(0.005)	−2.4 *
Work ability	-	-	0.03(0.01)	2.35 *
WFC*moderator ^1^	−0.01(0.005)	−2.50 *	-	-
FWC*moderator ^1^	0.01(0.006)	2.54 *	-	-
WHB*moderator ^1^	0.001(0.003)	0.39	−0.01(0.005)	−2.57 *
Marital status (no partner)	0.24(0.12)	2.11 *	0.28(0.11)	2.45 *
Parental status (no children)	−0.34(0.12)	−2.98 **	−0.26(0.11)	−2.34 *

*** = *p* < 0.001; ** = *p* < 0.01; * = *p* < 0.05; ^1^ In model 3, the moderator is age; in model 4, the moderator is work ability. Continuous variables in the interactions have been centered on the mean.

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
