# Peer review of "Work–Life Balance: Weighing the Importance of Work–Family and Work–Health Balance"

_ijerph, 2020, doi:10.3390/ijerph17030907_

Round 1
Reviewer 1 Report
This work is investigating the relationships between WHB, WLB, job satisfaction and other related variables. Although the research design with single-shot cross-sectional survey has a fundamental limitation (possible issues of common method bias), the research questions of this study and empirical data and analysis have scientific merit to contribute to the body of knowledge. And the sample size of +300 workers is not impressive. The authors seem to understand the limitation of the data and research design, but it should be more clearly stated for the readers of this paper. The authors should also clarify the samples. The authors stated that "The study involved workers of full age under an employment contract. Entrepreneurs and self-employed workers were excluded. We distributed the link to the online questionnaire with a brief description of the research through social networks, messaging applications, and email." but this is not enough. What kind of workers did you circulate your survey? What was the response rate?
And about the variables, the literature on job satisfaction and on work-life perceptions have identified numerous variables which are possible inter-related. To avoid issues of spurious relationship (other than the possible problem of common latent factor from single-method research), the authors should clarify how the research justify the lack of control variables in the model specification.
Going back to the issue of common variance due to the single-shot survey, the authors should state the result of Harman's single-factor test.
Author Response
REVIEWER 1
This work is investigating the relationships between WHB, WLB, job satisfaction and other related variables. Although the research design with single-shot cross-sectional survey has a fundamental limitation (possible issues of common method bias), the research questions of this study and empirical data and analysis have scientific merit to contribute to the body of knowledge.
We thank the reviewer for his judgment and agree with the limits indicated.
And the sample size of +300 workers is not impressive. The authors seem to understand the limitation of the data and research design, but it should be more clearly stated for the readers of this paper.
We added a clearer warning to the reader to not accept the results of the study without considering the limits. Line 575:
“The cited limitations warn against an unconditional generalization of the results of this study that, instead, have to be replicated with stronger research designs and other samples of workers.”
The authors should also clarify the samples. The authors stated that "The study involved workers of full age under an employment contract. Entrepreneurs and self-employed workers were excluded. We distributed the link to the online questionnaire with a brief description of the research through social networks, messaging applications, and email." but this is not enough. What kind of workers did you circulate your survey? What was the response rate?
We thank the reviewer for pointing out these relevant issues. As we stated in the procedure section, we used an online survey. As all survey modes, also web surveys have strengths and weaknesses. We circulated the survey on workers in the north-center of Italy. The specificity of our online recruitment made it very difficult to control the spread of the link to the questionnaire and it is impossible to accurately know which type of workers and how many visualized the invitation to the survey without deciding to participate. Therefore, it is impossible to calculate the proper response rate. However, we provided the completion rate at line 311 and specified the name of the social network media we used. We know this is a limit to the generalizability of our results and we discussed this issue in the limit section.
Moreover, as requested by the reviewer, we stated more clearly the limits of the study (line 575). However, in support of our results, we added a sentence on line 561
“In fact, as explained by Landers and Behrend [76], when the aim is to test theoretically relevant hypotheses, as in our study, sample representativeness is less crucial than when the study aims to estimate the presence and the level of one or more variables in the workers’ population. Of course, our results must be replicated in other samples to increase their generalizability.”
The same argument was used by the American Association for Public Opinion Research “There are times when a nonprobability online panel is an appropriate choice. […] Not all survey research is intended to produce precise estimates of population values. For example, a good deal of research is focused on improving our understanding of how personal characteristics interact with other survey variables such as attitudes, behaviors and intentions. Nonprobability online panels also have proven to be a valuable resource for methodological research of all kinds, as well as market research.” (Reg et al., 2010, pp. 48-49)
Our decision to adopt this kind of sample strategy was an informed decision (e.g., Reg et al., 2010; Fricker 2017; Landres & Behrend 2015). We evaluated the pros and cons of such online recruitment and we established that it was appropriate for this specific study. We hope that the reviewer will value our efforts.
Reg, B., Blumberg, S. J., Brick, J. M., Couper, M. P., Courtright, M., Dennis, J. M., … Lavrakas, P. J. (2010). Prepared for the aapor executive council by a task force operating under the auspices of the aapor standards committee, with members including. Public Opinion Quarterly, 74(4), 711–781. https://doi.org/10.1093/poq/nfq048]
Fricker, R. D. J. (2017). Sampling Methods for Online Surveys. In N. Fielding, R. M. Lee, & G. Blank (Eds.), The SAGE Handbook of Online Research Methods (pp. 162–183). Retrieved from https://search.ebscohost.com/login.aspx?direct=true&db=nlebk&AN=1447221&lang=it&site=eds-live&scope=site
Landers, R. N., & Behrend, T. S. (2015). An inconvenient truth: Arbitrary distinctions between organizational, mechanical turk, and other convenience samples. Industrial and Organizational Psychology, 8(2), 142–164. https://doi.org/10.1017/iop.2015.13
And about the variables, the literature on job satisfaction and on work-life perceptions have identified numerous variables which are possible inter-related. To avoid issues of spurious relationship (other than the possible problem of common latent factor from single-method research), the authors should clarify how the research justify the lack of control variables in the model specification.
We thank the reviewer for the suggestion. We decided not to include many other variables that were not involved in our specific hypotheses based on methodological recommendations that warn against the automatic inclusion of control variables in multiple regression (e.g., Breaugh, 2008; Spector & Brannick 2011; Becker et al. 2016). The authors of these recommendations highlighted many reasons why the inclusion of control variable should be carefully considered without believing in the so-called “purification principle”, “the implicit belief that statistical controls can yield more accurate estimates of relationships among variables of interest” (Spector & Brannick, 2011, p. 288). Moreover, Meade et al. (2009) specified that not including relevant variables is more problematic if the goal is to estimate precise path magnitudes and less troubling if significance or effect sizes is the goal, even when the omitted variable in question has a moderate to large correlation with the predictor. We based the inclusion of control variables (intervening or moderating) on strong literature, and we decided to include four moderators of the relation between work-family balance and Work Health Balance with job satisfaction: age, gender, parental status, and work ability. Of course, other variables could be added, but because only a few previous studies explored how the individual priorities moderate the effect of WLB on the outcomes (Casper et al., 2017; Sirgy & Lee, 2018), we believe that the study results are of value even if we did not consider all the possible variables. We hope the reviewer welcomes our arguments.
Breaugh, J. A. (2008). Important considerations in using statistical procedures to control for nuisance variables in non-experimental studies. Human Resource Management Review, 18(4), 282–293. https://doi.org/10.1016/j.hrmr.2008.03.001
Spector, P. E., & Brannick, M. T. (2011). Methodological Urban Legends: The Misuse of Statistical Control Variables. Organizational Research Methods, 14(2), 287–305. https://doi.org/10.1177/1094428110369842
Becker, T. E., Atinc, G., Breaugh, J. A., Carlson, K. D., Edwards, J. R., & Spector, P. E. (2016). Statistical control in correlational studies: 10 essential recommendations for organizational researchers. Journal of Organizational Behavior, 37(2), 157–167. https://doi.org/10.1002/job.2053
Meade, A. W., Behrend, T. S., & Lance, C. E. (2009). Dr. StrangeLOVE, or: How I learned to stop worrying and love omitted variables. In C. E. Lance & R. J. Vandenberg (Eds.), Statistical and methodological myths and urban legends: Doctrine, verity and fable in the organizational and social sciences (p. 89–106). Routledge/Taylor & Francis Group.
Going back to the issue of common variance due to the single-shot survey, the authors should state the result of Harman's single-factor test.
We thank the reviewer for pointing out the need for this test. We stated the result of Harman's single-factor test in the manuscript and provided the full EFA used for this analysis in the supplementary material. We added to line 348:
“Harman’s single-factor test was adopted to test the possibility of common method bias. The first factor explained 27% of the variance. Given the variance explained by the first factor falls below the threshold of 50%, common method bias does not appear to be a significant factor in this study. The results of the explorative factor analysis performed for the Harman single-factor test are available in the online supplementary material of this article.”

Reviewer 2 Report
This is a well-written paper to study importance of nonworking domains in work-life balance. Especially work-family/family-work balance is compared in importance with work-health balance regarding job satisfaction.
Manuscript is rather long, but anyhow logical. The theoretical background based on earlier literature is extensively described. Based on this, as well as to the lack of knowledge study questions are reasonable and clearly stated. Altogether there are 10 hypothesis to be tested. This of course makes it a bit difficult to follow in one reading, and the reader have to return to hypothesis formulation text while interpreting the results.
In material and methods (row 305-306) the ethical aspects of the research are described, but it is not mentioned about possible ethical committee evaluation of the research plan. If this was not done, an explanation for that would be needed.
There is no definition of what is meant by "have children", more explanation is required. How many? What age? Still at home? etc.. Another important issue might also be - especially among the older age group - if the participant had to take care of his/her parents with possible functional capacity problems. This, as one possible WFB issue in addition to having children should be taken in to account in the discussion section.
In the results section in tables the marks *, **, *** should have definitions.
The authors discuss the limitations of the study properly (cross-sectional, internet-based recruitment, selection bias, ad-hoc instrument ).
Also the study sample was rather small, and no strength calculations were done beforehand.
Some minor comments:
In row 137 "H1." should be in bold
In row 559 "et" in the beginning should be deleted
Author Response
REVIEWER 2
This is a well-written paper to study importance of nonworking domains in work-life balance. Especially work-family/family-work balance is compared in importance with work-health balance regarding job satisfaction.
Manuscript is rather long, but anyhow logical. The theoretical background based on earlier literature is extensively described. Based on this, as well as to the lack of knowledge study questions are reasonable and clearly stated. Altogether there are 10 hypothesis to be tested. This of course makes it a bit difficult to follow in one reading, and the reader have to return to hypothesis formulation text while interpreting the results.
We appreciate the reviewer's favorable opinion. We know the manuscript is quite long; we are hopeful that this will not discourage too many readers.
In material and methods (row 305-306) the ethical aspects of the research are described, but it is not mentioned about possible ethical committee evaluation of the research plan. If this was not done, an explanation for that would be needed.
We thank the reviewer for this remark. In fact, it was an oversight in the passage from the draft to the final version of the manuscript. In row 307, we placed the needed information:
“The study was conducted in accordance with the ethical standards set by the Declaration of Helsinki and was approved by the Ethical Committee of the University of Milano-Bicocca (Prot.160-2014).”
There is no definition of what is meant by "have children", more explanation is required. How many? What age? Still at home? etc..
We added in “Sample and Procedure” the information about the children. Line 315: “Among the 134 workers with children, 49%, 43%, and 8% of the respondents have one, two, and three or more children, respectively. The workers with one or more children under the age of twelve were 51%.”
Another important issue might also be - especially among the older age group - if the participant had to take care of his/her parents with possible functional capacity problems. This, as one possible WFB issue in addition to having children should be taken in to account in the discussion section.
We agree with the reviewer; it is likely to have the same increase in the effect of WFC on JS for workers responsible for eldercare. We added this issue in the discussion, line 514:
“Given this result, it is possible and should be tested in future studies if being responsible for eldercare, beyond generally increasing the level of work-to-family conflict, it also increases the impact of work-to-family conflict on job satisfaction.”
In the results section in tables the marks *, **, *** should have definitions.
We added the following line in the tables footer
“*** = p < 0.001; ** = p < 0.01; * = p < 0.05”
The authors discuss the limitations of the study properly (cross-sectional, internet-based recruitment, selection bias, ad-hoc instrument ).
Also the study sample was rather small, and no strength calculations were done beforehand.
Thanks for valuing our discussion about the limits of the study. We know the sample size was not “impressive”. We established the sample size based on the meta-analysis of Amstad et al. (2011) about the consequences of work-family conflict. The median sample size of the 98 studies selected in the review was 267. Based on this argument, we considered a minimum sample of 300 to be adequate. We are confident with our results even because the p value of all our models and many relevant estimates are lower than .005 (therefore quite far from the .05 threshold) and this reduces the likelihood of false-positive (Benjamin et al., 2018).
Amstad, F. T., Meier, L. L., Fasel, U., Elfering, A., & Semmer, N. K. (2011). A Meta-Analysis of Work-Family Conflict and Various Outcomes With a Special Emphasis on Cross-Domain Versus Matching-Domain Relations. Journal of Occupational Health Psychology, 16(2), 151–169. https://doi.org/10.1037/a0022170
Benjamin, D. J., Berger, J. O., Johannesson, M., Nosek, B. A., Wagenmakers, E. J., Berk, R., … Johnson, V. E. (2018). Redefine statistical significance. Nature Human Behaviour, 2(1), 6–10. https://doi.org/10.1038/s41562-017-0189-z
Some minor comments:
In row 137 "H1." should be in bold
We corrected.
In row 559 "et" in the beginning should be deleted
We thank the reviewer for pointing out this typo. We have deleted it.

Round 2
Reviewer 1 Report
The author(s) responded that the specificity of online recruitment made it very difficult to control the spread of the link to the questionnaire and it is impossible to accurately know which type of workers participated in the survey. This leaves a huge issue of selection problems. Every data has limitation, but at least we should know the limitation. Readers are blind about the sampled workers/respondents, so we don't even conclude whether we can accept this research results or not. Too arbitrary sampling.
Author Response
"Please see the attachment."
